# HER2 Expression in Peritoneal Dissemination of High-Grade Serous Ovarian Carcinoma: A Comparative Study of Immunohistochemical Reactivity Using Four HER2 Antibodies

**DOI:** 10.3390/jcm11236963

**Published:** 2022-11-25

**Authors:** Min-Kyung Yeo, Sup Kim, Heon Jong Yoo, Kwang-Sun Suh, Kyung-Hee Kim

**Affiliations:** 1Department of Pathology, Chungnam National University School of Medicine, Chungnam National University Hospital, Daejeon 35015, Republic of Korea; 2Department of Radiation Oncology, Chungnam National University School of Medicine, Chungnam National University Hospital, Daejeon 35015, Republic of Korea; 3Department of Obstetrics and Gynecology, Chungnam National University School of Medicine, Daejeon 35015, Republic of Korea; 4Department of Obstetrics and Gynecology, Chungnam National University Sejong Hospital, Sejong-si 30099, Republic of Korea; 5Department of Pathology, Chungnam National University Sejong Hospital, Sejong-si 30099, Republic of Korea

**Keywords:** HER2, serous carcinoma, ovary, peritoneum, immunohistochemistry, antibodies

## Abstract

Most high-grade serous ovarian carcinomas (HGSOCs) involving the peritoneum are aggressive. Epidermal growth factor receptor 2 (HER2) is aberrantly activated in a variety of solid cancers. The HER2 status of a tumor is based on cytoplasmic membrane staining of an intracellular domain (ICD)-specific HER2 antibody. We compared four anti-HER2 antibodies in an immunohistochemical study of HGSOC with peritoneal dissemination. HER2 expression was assessed in peritoneal disseminated HGSOC specimens from 38 patients by immunohistochemistry using four different anti-HER2 antibodies (an ICD antibody (clone A0485), an extracellular domain (ECD) antibody (clone SP3), and two antibodies recognizing HER2 phosphorylated at tyrosine 877 or 1248 (pHER2Y^877^ and pHER2Y^1248^)). *HER2* gene amplification was accessed by chromogenic in situ hybridization (CISH). The antibodies showed HER2 positivity as follows: 31.6% of cases (12/38) with A0485, 26.3% (10/38) with SP3, 7.9% (3/38) with pHER2Y^877^, and 21.1% (8/38) with pHER2Y^1248^. Fifteen out of thirty-eight (39.5%) cases were positive for at least one of the four HER2 antibodies. *HER2* gene amplification was detected in 3/19 cases. All four HER2 antibodies could be used for patient selection for anti-HER2 therapies. These findings raise the possibility of anti-HER2 therapeutic strategies for HGSOC with peritoneal dissemination.

## 1. Introduction

High-grade serous ovarian carcinoma (HGSOC) arises from tubal-type epithelium in the fallopian fimbria, ovarian surface, or ovarian epithelial inclusion cysts, and commonly harbors *TP53* mutations [1]. The vast majority of patients with HGSOC have omental involvement at diagnosis. The mortality of HGSOC is high, accounting for >70% of ovarian cancer deaths [2,3]. Platinum-based chemotherapy is a cornerstone of HGSOC therapy, and >80% of patients achieve a primary response; however, most patients will relapse and develop resistance to platinum-based therapies [4,5].

Epidermal growth factor receptor 2 (HER2) amplification occurs in many types of cancer, including breast, gastric, colon, bladder, and biliary cancers, but anti-HER2 therapy in HGSOC has not yet been fully evaluated [6]. Clinically, HER2 expression status, as determined by cytoplasmic membrane expression of the intracellular domain (ICD) of HER2 and/or *HER2* gene amplification, is a major factor in determining the use of anti-HER2 therapy. The U.S. Food and Drug Administration (FDA)-approved companion diagnostics for determining the use of anti-HER2 therapies use ICD-targeting HER2 antibodies (https://www.fda.gov/medical-devices/in-vitro-diagnostics/list-cleared-or-approved-companion-diagnostic-devices-in-vitro-and-imaging-tools accessed on 1 November 2022). However, both ICD-targeting anti-HER2 therapies (e.g., neratinib, lapatinib, afatinib) and extracellular domain (ECD)-targeting antibodies (e.g., trastuzumab, pertuzumab, T-DM1) are FDA-approved [7].

HER2 dimerization with other HER2 family members (EGFR/HER1, HER3, or HER4) leads to phosphorylation of specific residues within the ICD of HER2 and initiates downstream signaling [8,9]. In a previous study, increased phosphorylation of HER2 at tyrosine 1248 (pHER2Y^1248^) was an independent marker for poor clinical outcomes, including disease-free survival (DFS) and overall survival (OS), in breast cancer patients [10]. On the other hand, increased expression of pHER2Y^1248^ showed a positive correlation with the response to trastuzumab in HER2 ICD-positive breast cancers [11,12]. Almost 5% of HER2 ICD-negative breast cancer cases showed increased HER2 phosphorylation at tyrosine 877 (pHER2Y^877^), and trastuzumab has been shown to reduce the proliferation of HER2 ICD-negative/pHER2Y^877^-positive breast cancer cell lines [13]. Therefore, additional assessment of HER2 phosphorylation status may be more effective in determining whether a patient is a suitable candidate for anti-HER2 therapy than simple measurement of HER2 ICD expression or *HER2* gene amplification.

To our knowledge, there is no study focusing on evaluation of HER2 phosphorylation status in HGSOC. Investigation of the frequency of HER2 ECD and pHER2 expression could validate the use of HER2-targeted therapies in patients with peritoneal disseminated HGSOC. This study therefore evaluated the expression of HER2 ICD, HER2 ECD, and pHER2 in disseminated peritoneal cancer cells of HGSOC, as well as associations with the clinical and pathological characteristics of HGSOC.

## 2. Materials and Methods

### 2.1. Patient and Tissue Samples

This study was approved by the Institutional Review Board of Chungnam National University Hospital (CNUH 2019-10-041). The requirement for informed consent was waived because this was a retrospective immunohistochemical study and dual-color chromogenic in situ hybridization (CISH) study that used formalin-fixed, paraffin-embedded (FFPE) tissue. Specimens were collected from 38 patients with peritoneal disseminated HGSOC who underwent primary surgical resection between 2011 and 2017 at Chungnam National University Hospital in Daejeon, South Korea. Representative FFPE tissue samples from peritoneal lesions of HGSOC were analyzed.

HGSOCs involving one or both ovaries or fallopian tubes and with peritoneal extension were included. The patients underwent surgical excision, and the tumors were classified as pathologic tumor stage 2 (pT2) or 3 (pT3) according to the American Joint Committee on Cancer (AJCC) classification [14]. The exclusion criteria related to limiting the interpretation of HER2 expression in study results were as follows: (1) patients had a previous history of other cancers; (2) patients had received previous curative resection for any ovarian tumor lesion; and (3) patients had received any chemotherapy or radiotherapy.

The pathologic tumor, node, and metastasis (pTNM) stage and histologic grade of the HGSOCs were determined at the time of surgical resection and were based on the 8th edition of the AJCC staging system [14].

### 2.2. Immunohistochemical Staining and Analysis

Immunohistochemical staining was conducted as previously described [15]. Target Retrieval Solution, pH 9 (catalog #S2368; Dako, Glostrup, Denmark), was used for antigen revitalization. Peroxide blocking was performed using 0.3% H_2_O_2_ at room temperature for 10 min. The tissue sections were incubated at 37 °C for 30 min with the following primary antibodies: rabbit polyclonal anti-human c-erbB2 antibody (clone A0485, 1:300; Dako, Glostrup, Denmark), rabbit monoclonal anti-erbB2 antibody (clone SP3, 1:100, ab16662; abcam, Cambridge, UK), rabbit monoclonal anti-HER2 (phospho Y877) antibody (clone EP2324Y, 1:100, ab108371; Abcam, Cambridge, UK), and rabbit polyclonal anti-HER2 (phospho Y1248) antibody (1:100, ab227769; abcam). For detection, the EnVision+ System-HRP was used with Dako EnVision+ System-HRP anti-rabbit labeled polymer and DAB+ substrate chromogen (codes K4003 and K3468; Dako, Glostrup, Denmark).

We scored membranous immunoreactivity for HER2 protein expressions [16]. A staining score for each of the four HER2 antibodies was assigned as follows: 0, no reactivity or membranous reactivity in <10% of tumor cells; 1, faint perceptible membranous reactivity in ≥10% of tumor cells, visible at 200×; 2, weak to moderate complete basolateral or lateral membranous reactivity in ≥10% of tumor cells, visible at 100×; and 3, strong complete basolateral or lateral membranous reactivity in ≥10% of tumor cells, visible at 40×. A score of 3 was regarded as a positive expression. The stained slides were examined separately and scored by Kim, K.-H., and Yeo, M.-K., who were blinded to the patients’ clinicopathological details. Any discrepancies in the scores were discussed to obtain a consensus.

### 2.3. Dual-Color Chromogenic In Situ Hybridization (dc-CISH)

FFPE tissue sections were analyzed by dc-CISH using the ZytoDot 2C SPEC ERBB2/D17S122 Probe kit (C-3068-100; ZytoVision, Bremerhaven, Germany), which is designed for the detection of *HER2* gene amplification, according to manufacturer instructions [17]. *HER2* is located in the chromosomal region 17q12. The SPEC D17S122 probe is designed to detect the chromosome 17 copy number instead of the alpha satellite centromeric region of chromosome 17, where gains or losses can occur.

Interpretation of dc-CISH was performed separately by K.-H. Kim and M.-K. Yeo with a 60X objective. At least 60 nuclei from malignant cells with both green (HER2) and red (D17S122) signals were scored per section. Positivity for *HER2* gene amplification was defined as ≥6 green signals per tumor nucleus in >50% of cancer cells [17,18].

### 2.4. Statistical Analyses

Associations between immunohistochemical detection of HER2 ICD, HER2 ECD, pHER2Y^877^, and pHER2Y^1248^, and clinicopathological parameters were evaluated by Pearson’s chi-square tests and Fisher’s exact tests. Differences in HER2 expression with the four antibodies were assessed using Wilcoxon signed-rank tests and McNemar’s tests. Correlations between HER2 protein expression determined using different antibodies were assessed using Spearman’s correlations. Statistical significance was set at *p* < 0.05 (SPSS v.26; SPSS Inc., Chicago, IL, USA).

## 3. Results

### 3.1. Relationships between HER2 Expression and Clinicopathological Characteristics

Clinicopathological characteristics are shown Table 1. The median age of the study population was 57 years, and the range was 39–87 years. Complete or incomplete surgical resection was achieved in all 38 patients. No neoadjuvant chemotherapy and/or radiotherapy was administered to enrolled patients.

HER2 expression was evaluated in 38 peritoneal disseminated HGSOC specimens using the four anti-HER2 antibodies. All four anti-HER2 antibodies showed positive staining with circumferential, complete basolateral, or lateral membranous reactivity. The HER2 positivity by the four assays showed no statistical correlation with any clinicopathologic characteristic evaluated, including age, pT stage, or pN stage.

Disease free survival (DFS) was defined as the period between initial surgical resection of HGSOC involving the peritoneum and the date of HGSOC recurrence or metastasis. Overall survival (OS) was defined as the period from the date of initial surgery to the date of death or last follow-up. The mean follow-up durations for OS and DFS were 71.46 ± 9.712 and 30.61 ± 6.150 months. Both OS and DFS analyses were performed for the 38 patients. The HER2 positivity was defined as positive for at least one of the four HER2 antibodies or positive HER2 gene amplification (HER2 negative 22 cases, HER2 positive 16cases). Kaplan–Meier survival curves and log-rank tests for HER2 expression showed no significant association with DFS (log-rank = 1.355, *p* = 0.244) or OS (log-rank = 0.380, *p* = 0.538), although the 16 HER2 positive patients had a shorter DFS period (Figure 1).

### 3.2. Comparing HER2 Expression with the Four Anti-HER2 Antibodies in Peritoneal Disseminated HGSOC

In total, 39.5% (15/38) of peritoneal disseminated HGSOCs showed HER2 positivity with at least one of the four anti-HER2 antibodies, while only two cases tested positive with all four anti-HER2 antibodies (Figure 2). Staining with A0485, the HER2 ICD-specific antibody, was positive in 31.6% of cases (12/38). By contrast, staining for pHER2Y^877^ was positive in 7.9% of cases (3/38) (Figure 3). Significant differences were observed between the pHER2Y^877^ assay and the A0485 or SP3 assays (*p* = 0.012 and *p* = 0.039, respectively) (Table 2).

### 3.3. HER2 Gene Amplification by Dual-Color Chromogenic In Situ Hybridization

The CISH test was performed on samples with at least one of the four anti-HER2 antibody immunostaining scores of 3 or at least three antibody scores of 2. Twenty one out of 38 cases were assessed, and 2 cases failed. These data indicate that immunohistochemical study using various anti-HER2 antibodies may contribute to determine whether a patient is eligible for HER2-targeted therapies Positive *HER2* gene amplification was seen in 3/19 of the HGSOC specimens: case 22, case 27, and case 29 (Figure 1 and Figure 3). Case 22 showed HER2 positivity using all four anti-HER2 antibodies (A0485, SP3, pHER2Y^877^, and pHER2Y^1248^), while case 27 was negative for HER2 expression using all four anti-HER2 antibodies (score of 2 for all four assays), and case 29 was positive using three of the anti-HER2 antibodies (A0485, SP3, and pHER2Y^1248^). Cases 22 and 29 had a HER2 expression score of 3, with complete cytoplasmic membranous reactivity, using the A0485 and SP3 antibodies (Figure 2).

**Figure 3 jcm-11-06963-f003:**
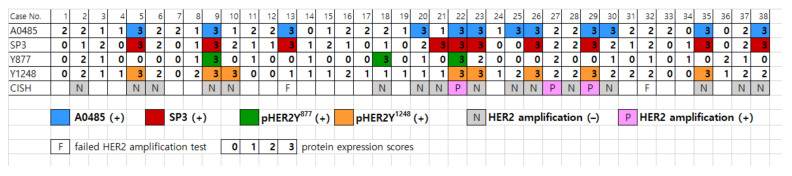
Positivity (score 3) for the four HER2 antibodies (A0485, SP3, pHER2Y^877^, and pHER2Y^1248^) and *HER2* amplification by CISH in peritoneal disseminated cancer cells of HGSOC.

## 4. Discussion

We hypothesized that HER2 expression might be involved in the peritoneal dissemination of HGOSC, suggesting that anti-HER2 targeted therapy could be administered in the adjunct setting. We evaluated HER2 expression in 38 patients with peritoneal disseminated HGSOC using four different anti-HER2 antibodies: A0485 (ICD), SP3 (ECD), pHER2Y^877^, and pHER2Y^1248^. Additionally, *HER2* amplification status was evaluated by ds-CISH. We found HER2 positivity with at least one of the four anti-HER2 antibodies in 39.5% of cases (15/38). Specifically, 31.6% of cases (12/38) showed positive staining with A0485, 26.3% of cases (10/38) showed positive staining with SP3, 7.9% cases (3/38) were positive with pHER2Y^877^, and 21.1% of cases (8/38) were positive with pHER2Y^1248^. Only 2 of 38 HGSOCs showed positive staining with all four anti-HER2 antibodies, and significant discrepancies between the pHER2Y^877^ assay and the A0485 or SP3 assays were observed. Presently, HER2 ICD expression status with A0485 (Dako) or 4B5 (Ventana, Roche Diagnostics.) and *HER2* gene amplification status are FDA-approved companion diagnostic tests for HER2-targeted therapy [6]. Both A0485 and PATHWAY anti-Her2/neu (4B5) were selected as FDA-approved companion diagnostics for breast cancer. Previous study results have shown that A0485, HercepTest, SP3 and 4B5 were over 95% sensitive and specific in 184 cases of breast cancer [19]. However, little is known about the HER2 test using 4B5 antibody in ovarian cancers, and only a few studies of ovarian clear cell carcinoma have been reported [20,21]. In one study of 71 ovarian clear cell carcinoma cases, the immunohistochemical staining concordance rates between A0485 and 4B5 ranged from 87.3 to 93.0%. [20]. Future studies are needed to determine the concordance between A0485 and 4B5 immunohistochemical staining results in ovarian serous carcinoma.

The focus of FDA-approved HER ICD assays may limit the use of HER2-targeted therapy in patients with overexpression of the HER2 ECD or phosphorylated HER2 [22,23,24]. These data indicate that immunohistochemical study using various anti-HER2 antibodies may contribute to determine whether a patient is eligible for HER2-targeted therapies. Several studies have showed that, in addition to *HER2* gene amplification, *HER2* gene mutations are related to HER2 expression status [25,26,27,28]. Thus, *HER2* gene amplification, *HER2* gene mutation, and HER2 protein expression should be defined in the context of eligibility for various anti-HER2 targeted therapies [6,29].

Recently, heterogenous expression of HER2 in breast and gastric cancer has emerged as an important factor for HER2-targeted therapy [30,31]. For this reason, it has been proposed to evaluate HER2 status using multiple tissue blocks to improve the HER2 positivity rate in gastric and breast cancer [32,33]. In addition, recent studies have shown significant efficacy of HER2-targeted agents for advanced breast cancer patients with low HER2 expression (IHC score of 1+ or 2+ without HER2 gene amplification), suggesting that breast cancer patients with low HER2 expression may also be selected for HER2-targeted therapy [31,34].

HER2 signaling is known to lead to carcinogenesis and tumor progression in various cancers, including breast, gastric, and lung carcinomas [35]. Most previous studies that have performed immunohistochemical staining for HER2 in ovarian serous carcinoma showed rare positive HER2 expression, and aberrant HER2 activation in HGSOC has not yet been evaluated, particularly with respect to the potential application of HER2-targeted therapy [35,36,37,38].

HGSOC is the most common epithelial ovarian cancer and has the highest mortality rate among ovarian carcinomas, accounting for 70–80% of ovarian cancer deaths [39]. Most cases of peritoneal dissemination of HGSOC are superficially invasive in the peritoneal cavity, but regional lymph nodes or visceral organs can be involved [40,41]. Even in the absence of lymphovascular invasion, HGSOC cells primarily spread into the peritoneal cavity and passively expand, with a preference for adipocytes [2]. Upregulation of fatty acid-binding protein 4 (FABP4), also known as adipocyte protein 2, has been found in omental metastases of serous ovarian cancer compared with primary ovarian cancers [2]. In breast cancer, FABP4 expression was highest in the HER2 subtype and lowest in the luminal A subtype [42], and HER2 overexpression-mediated oncogenic transformation of breast cells was accompanied by increased FABP4 expression [43]. HER2 in peritoneal disseminated HGSOC may also be involved in seeding the cancer cells in the peritoneal adipose tissue.

In a previous study of 50 HGSOC cases, including primary sites and metastases, 52% (26/50 cases) showed some HER2 expression, with scores ranging from 1 to 3, but only one case had a HER2 expression score of 3+ [38]. In another study, HER2 3+ expression was seen in only 1 out of 53 cases of primary serous carcinoma, and *HER2* amplification by CISH was not identified in 43 serous carcinomas (0/43), so the authors concluded that HER2 was not altered in ovarian serous carcinoma [35]. In a study of 56 ovarian serous carcinomas, all were negative for HER2 reactivity, while 18.9% (7/37 cases) of uterine serous carcinomas were positive [37]. In another primary uterine and ovarian serous carcinoma study, HER2 expression using both ICD (CB11) and ECD (SP3) antibodies was higher in uterine serous carcinoma than in ovarian serous carcinoma, while high ICD staining with low ECD staining was seen in 8% (6/75) of uterine serous carcinoma cases and 42% (29/69) of ovarian serous carcinoma cases [7]. However, in our analysis of peritoneal disseminated HGSOCs, 24% of cases (9/38) were positive for staining with both the HER2 ICD (A0485) and HER2 ECD (SP3) antibodies, and 8% of cases (3/38) were positive with the ICD antibody and negative with the ECD antibody. These results suggest the potential benefit of combining HER2 ICD-targeted therapies (e.g., lapatinib or afatinib) with ECD-directed therapies (e.g., trastuzumab, pertuzumab, or T-DM1) in peritoneal disseminated HGSOC. Various tyrosine kinase inhibitors (TKIs) target the HER2 and act by inhibiting HER2 receptor signaling in multiple cancers. These TKIs include lapatinib, canertinib, neratinib, sapitinib, and dacomitinib [44]. Preclinical studies of HER2-targeted TKIs in ovarian cancer patients have been evaluated. Clinical trials (phase I or II) combining gefitinib with cisplatin, erlotinib with carboplatin, and sapitinib with paclitaxel showed improved clinical efficacy [45,46,47]. Further studies of the HER2-targeted TKI therapies are required to demonstrate the contribution of TKIs in these combinations.

In a phase 2 clinical trial of trastuzumab in 41 patients with recurrent or refractory ovarian or primary peritoneal carcinoma and HER2 expression scores of 2 or 3, the overall objective response rate was 7.3% (one complete response and two partial responses) [48]. Treatment with pertuzumab in 117 advanced recurrent epithelial ovarian patients who had previously received a platinum-based chemotherapy resulted in five partial responses (response rate: 4.3%). Of the 117 patients, the pHER2 level was assessed in 28 pre-treatment tumor samples by enzyme-linked immunosorbent assay, and 8 of the 28 were pHER2-positive. The eight pHER2-positive patients had better responses to pertuzumab than the pHER2-negative patients, with a median progression-free survival of 20.9 weeks for pHER2-positive patients and 5.8 weeks for pHER2-negative patients. No tumor biopsies were found to have *HER2* gene amplification. A partial response was achieved in one of the eight patients with pHER2-positive tumors (12.5%) but in none of the 20 patients with pHER2-negative tumors. Log-rank testing for the pertuzumab efficacy showed a trend toward improved survival in patients with pHER2-positive tumors compared with patients with pHER2-negative tumors. Pertuzumab efficacy may therefore be higher in pHER2-positive tumors [8]. In breast tumor models without HER2 overexpression or *HER2* amplification, pertuzumab has been shown to inhibit ligand-stimulated signaling by interfering with binding to HER3 [49].

Previous reports have investigated HER2 expression status in ovarian lesions of HGSOC or regardless of tumor location, but this study focused on peritoneal lesions of HGSOC. HER2 ECD overexpression, HER2 phosphorylation, or various HER2 gene mutations, as well as HER2 ICD overexpression or *HER2* gene amplification can lead to HER2 signaling pathway activation. We investigated HER2 status using four different anti-HER2 antibodies. In total, 42.1% (16/38) of peritoneal disseminated HGSOCs showed HER2 positivity with at least one of the four anti-HER2 antibodies or positive HER2 gene amplification.

Our results show different HER2 positivity rates by immunohistochemistry in 38 cases of peritoneal disseminated HGOSC using four different anti-HER2 antibodies [A0485 (ICD-binding), SP3 (ECD-binding), pHER2Y^877^, and pHER2Y^1248^]. These findings bring considerable interest to HER2-targeted therapy of HGSOC. Measurement of HER2 levels using various HER2 antibodies, including phosphorylated HER2 antibodies, could increase our recognition of activating HER2 mutations and lead to broader use of HER2-targeted therapies. HER2 evaluation in multiple tissue blocks is recommended to accurately assess intratumoral HER2 heterogeneity. Immunohistochemical studies using different HER2 antibodies in multiple tissue blocks can be more cost-effective than in situ hybridization methods to detect *HER2* gene amplification in multiple blocks. Future research should evaluate the HER2 positivity using both extracellular and intracellular domain-specific HER2 antibodies, as well as antibodies specific for phosphorylated HER2, to assess the potential use of anti-HER2 therapeutic strategies. We suggest that HER2 might be involved in peritoneal dissemination of HGOSC and could be a promising therapeutic target.

## 5. Conclusions

According to this immunohistochemical analysis using four different anti-HER2 antibodies (antibodies against pHER2Y^877^ and pHER2Y^1248^, as well as HER2 ICD- and ECD-specific antibodies), HER2 expression in peritoneal disseminated HGSOC was diverse. The differential expression of HER2 could be used to determine a personalized therapeutic strategy with HER2-targeted therapies. HER2-targeted therapies have become more broadly used recently, although the possible role of HER2 in HGSOC has not been elucidated. Therefore, further mechanistic and clinical studies on the efficacy of various HER2-targeted therapies based on both HER2 expression pattern and HER2 phosphorylation status are needed.

## Figures and Tables

**Figure 1 jcm-11-06963-f001:**
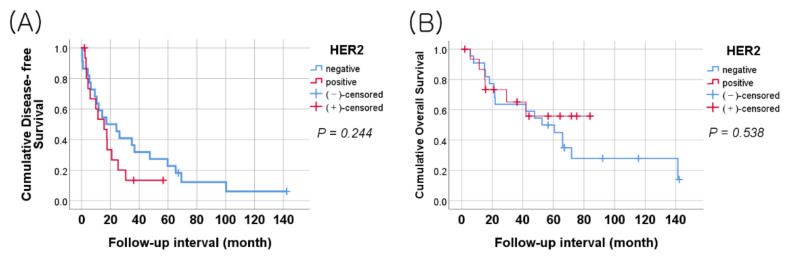
Kaplan–Meier survival curves. (**A**) Disease free survival (DFS) curve in 38 patients with HER2 expression. (**B**) Overall survival (OS) curve in 38 patients with HER2 expression. The HER2 status showed no statistically significant relationship with DFS or OS.

**Figure 2 jcm-11-06963-f002:**
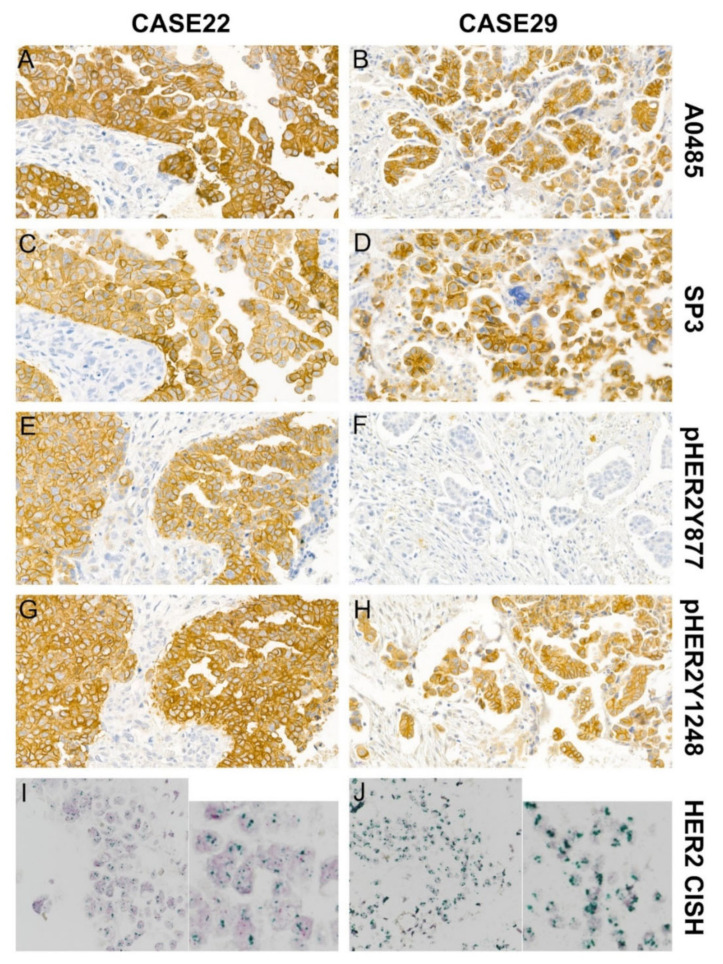
*HER2* CISH and HER2 immunohistochemical staining in peritoneal disseminated cancer cells of HGSOC using four anti-HER2 antibodies: A0485, SP3, pHER2Y^877^, and pHER2Y^1248^ (**A**–**J**). Cases 22 and 29 showed positive HER2 amplification (**I**,**J**). Original magnification, 600×. Case 22 shows positive HER2 expression using all four anti-HER2 antibodies (**A**,**C**,**E**,**G**), and case 29 was positive using three antibodies (A0485, SP3, and pHER2Y^1248^) (**B**,**D**,**H**). Original magnification, 400×.

**Table 1 jcm-11-06963-t001:** Clinicopathological characteristics associated with positivity in the four HER2 assays.

	A0485			SP3			pHER2Y^877^			pHER2Y^1248^		
	(−)	(+)	*p* *	(−)	(+)	*p* *	(−)	(+)	*p* *	(−)	(+)	*p* *
Case no.	26	12		28	10		35	3		30	8	
Age			0.337 **			0.460			0.081			0.709
<60	13	8		14	7		21	0		16	5	
≥60	13	4		14	3		14	3		14	3	
pT stage			0.270			0.709			0.538			1.000
2	7	6		9	4		13	0		10	3	
3	19	6		19	6		22	3		20	5	
pNode			0.395			0.156			1.000			0.307
(−)	20	11		21	10		28	3		23	8	
(+)	6	1		7	0		7	0		7	0	
pTNM stage			0.270 ***			0.709 ***			0.538 ***			1.000 ***
I	0	0		0	0		0	0		0	0	
II	7	6		9	4		13	0		10	3	
III	17	5		16	6		19	3		17	5	
IV	2	1		3	0		3	0		3	0	

*, Pearson’s chi-squared test; **, Fisher’s exact test; Node, lymph node metastasis; ***, Fisher’s exact test (stage I and II versus III and IV); TNM, tumor-node-metastasis.

**Table 2 jcm-11-06963-t002:** Correlations among positivity in the four HER2 assays.

	A0485		
Matched Pairs	Negative (26 cases)	Positive (12 cases)	*p* *
SP3 (no.)			0.625
negative	25	3	
positive	1	9	
pHER2Y^877^			0.012
negative	25	10	
positive	1	2	
pHER2Y^1248^			0.219
negative	25	5	
positive	1	7	

*, McNemar’s test.

## Data Availability

Supporting data may be found in internal archives.

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
