# Peer review of "HER2 Expression in Peritoneal Dissemination of High-Grade Serous Ovarian Carcinoma: A Comparative Study of Immunohistochemical Reactivity Using Four HER2 Antibodies"

_jcm, 2022, doi:10.3390/jcm11236963_

Round 1
Reviewer 1 Report
This manuscript evaluates the staining patterns of 4 distinct anti-HER2 antibodies, including 1 FDA-approved diagnostic and 2 that are phospho-HER2-specific, in 38 treatment-naive peritoneal-disseminated HGSOC. The presence of HER2 amplification was assessed in 19 of these samples, as well. This manuscript exemplifies the heterogeneity in HER2 detection in HGSOC, and provides valuable information to the field. However, there are some concerns which should be addressed:
1) It would be helpful to conduct a more thorough literature search on TKI inhibitor use and outcomes in HGSOC, such as PMID 15684311 and reviewed in 30766749.
2) The provided FDA link is not functional.
3) A0485 was selected as the representative FDA-approved diagnostic, but it is unclear whether the 4B5 (PATHWAY) antibody would show equivalent positivity in these patient samples. If it is not possible to stain with this antibody, at minimum, the literature-reported general staining differences between these two antibodies should be discussed.
4) It is unclear why only 19 of the samples were assessed for HER2 amplification, instead of the entire cohort?
5) The discussion claim “These data suggest that various anti-HER2 antibodies in addition to an ICD-specific antibody should be used to determine whether a patient is eligible for HER2-targeted therapies.” Is unfounded based on the presented data. A large-scale experiment would be necessary to determine the relationship of staining with treatment outcomes.
5) In the conclusion, "HER2 expression patterns" are mentioned, however phosphorylation status may also be important. These statements should be updated to include both expression and phosphorylation.
Author Response
1) It would be helpful to conduct a more thorough literature search on TKI inhibitor use and outcomes in HGSOC, such as PMID 15684311 and reviewed in 30766749.
: In the revised version of the manuscript, we correct as as shown below. (line 309-315)
Various tyrosine kinase inhibitors (TKIs) target the HER2 and act by inhibiting HER2 receptor signaling in multiple cancers. These TKIs include lapatinib, canertinib, nerat-inib, sapitinib, and dacomitinib [30766749]. Preclinical studies of HER2-targeted TKIs in ovarian cancer patients have been evaluated. Clinical trials (phase I or II) combining gefitinib with cisplatin, erlotinib with carboplatin, and sapitinib with paclitaxel showed improved clinical efficacy (PMID: 22313686, PMID: 20646751, PMID: 23589215). Further studies of the HER2-targeted TKI therapies are required to demonstrate the contribution of TKIs in these combinations.
2) The provided FDA link is not functional.
In the revised version of the manuscript, the FDA link will be activated with Ctrl press. (line 53-54)
3) A0485 was selected as the representative FDA-approved diagnostic, but it is unclear whether the 4B5 (PATHWAY) antibody would show equivalent positivity in these patient samples. If it is not possible to stain with this antibody, at minimum, the literature-reported general staining differences between these two antibodies should be discussed.
: Both A0485 and PATHWAY anti-Her2/neu (4B5) were selected as FDA-approved companion diagnostics for breast cancer. Previous study results have shown that A0485, HercepTest, SP3 and 4B5 were over 95% sensitive and specific in 184 cases of breast cancer (PMID: 23867547). However, little is known about the HER2 test using 4B5 antibody in ovarian cancers, and only a few studies of ovarian clear cell carcinoma have been reported (PMID: 34036622 PMID: 29989198). In one study of 71 ovarian clear cell carcinoma cases, the immunohistochemical staining concordance rates between A0485 and 4B5 ranged from 87.3 to 93.0%. (PMID: 34036622). Future studies are needed to determine the concordance between A0485 and 4B5 immunohistochemical staining results in ovarian serous carcinoma.
: In the revised version of the manuscript, we have inserted the above sentences (line241-249)
4) It is unclear why only 19 of the samples were assessed for HER2 amplification, instead of the entire cohort?
The CISH test was performed on samples with at least one of the four anti-HER2 antibody immunostaining scores of 3 or at least three antibody scores of 2. Twenty one out of 38 cases were assessed, and 2 cases failed.
: In the revised version of the manuscript, we inserted the sentence above and modified Figure 3 (line 201-205, Fig. 3).
5) The discussion claim “These data suggest that various anti-HER2 antibodies in addition to an ICD-specific antibody should be used to determine whether a patient is eligible for HER2-targeted therapies.” Is unfounded based on the presented data. A large-scale experiment would be necessary to determine the relationship of staining with treatment outcomes.
: In the revised version of the manuscript, we correct as shown below. (line 252)
: These data indicate that immunohistochemical study using various anti-HER2 antibodies may contribute to determine whether a patient is eligible for HER2-targeted therapies.
6 In the conclusion, "HER2 expression patterns" are mentioned, however phosphorylation status may also be important. These statements should be updated to include both expression and phosphorylation.
Therefore, further mechanistic and clinical studies on the efficacy of various HER2-targeted therapies based on both HER2 expression pattern and HER2 phosphorylation status are needed.
: In the revised version of the manuscript, we have inserted “HER2 phosphorylation status” (line 367).
Reviewer 2 Report
First, I would like to congratulate the authors on their work regarding HER2 expression in peritoneal dissemination of HGSOC. I see these tumors on a regular basis in my clinical practice and being aware of their aggressiveness, I strongly believe that the introduction of any novel targeted therapies which may emerge from these kinds of studies are extremely valuable.
Therefore, I believe the subject is extremely appealing and the manuscript appears well written. However, I have a few suggestions for the authors:
1. Please review the presence of reference [14] on line 89. I believe it should be replaced with “according to AJCC classification…” or something similar, instead of the reference number.
2. It might be appropriate to mention somewhere in the materials and methods section the reasoning for exclusion criteria.
3. I believe it would be important to disclose if the pathologists who examined the immunoreactivity for HER2 have experience in evaluating HER2 on breast (or gastric, etc.) specimens.
4. Can the authors provide any mentions regarding the survival/follow-up in regards to HER2 expression?
5. Lines 207-212 require references. Please ensure it’s clear which “several studies”.
6. Lines 235-236 please specify primary serous carcinoma of which organ.
7. The study findings are extremely interesting. Could you add a short paragraph commenting the differences from the results published in the literature?
8. Lines 275-276: can the authors comment on the cost-effectiveness of using multiple HER2 antibodies in clinical practice versus CISH.
Author Response
1. Please review the presence of reference [14] on line 89. I believe it should be replaced with “according to AJCC classification…” or something similar, instead of the reference number.
: Thank you for your comments which we have addressed on line 90 of the revised manuscript.
2. It might be appropriate to mention somewhere in the materials and methods section the reasoning for exclusion criteria.
: Thank you for your comments which we have addressed on line 91 and 92 of the revised manuscript as shown below.
The exclusion criteria related to limiting the interpretation of HER2 expression in study results were as follows: (1) patients had a previous history of other cancers; (2) patients had received previous curative resection for any ovarian tumor lesion; and (3) patients had received any chemotherapy or radiotherapy.
3. I believe it would be important to disclose if the pathologists who examined the immunoreactivity for HER2 have experience in evaluating HER2 on breast (or gastric, etc.) specimens.
: Recently, heterogenous expression of HER2 in breast and gastric cancer has emerged as an important factor for HER2-targeted therapy (PMID34519765, PMID33010699). For this reason, it has been proposed to evaluate HER2 status using multiple tissue blocks to improve the HER2 positivity rate in gastric and breast cancer (PMID: 35765007 PMID: 30653031). In addition, recent studies have shown significant efficacy of HER2-targeted agents for advanced breast cancer patients with low HER2 expression (IHC score of 1+ or 2+ without HER2 gene amplification), suggesting that breast cancer patients with low HER2 expression may also be selected for HER2-targeted therapy (PMID: 31821109, PMID33010699).
: In the revised version of the manuscript, we have inserted the above sentences (line 258-265)
4. Can the authors provide any mentions regarding the survival/follow-up in regards to HER2 expression?
: We have inserted data correlating recurrence/survival outcome with HER2 expression as shown below. (line 162-172. Figure 1.)
Disease free survival (DFS) was defined as the period between initial surgical resection of HGSOC involving the peritoneum and the date of HGSOC recurrence or metastasis. Overall survival (OS) was defined as the period from the date of initial surgery to the date of death or last follow-up. The mean follow-up durations for OS and DFS were 71.46±9.712 and 30.61±6.150 months. Both OS and DFS analyses were performed for the 38 patients. The HER2 positivity was defined as positive for at least one of the four HER2 antibodies or positive HER2 gene amplification (HER2 negative 22 cases, HER2 positive 16cases). Kaplan-Meier survival curves and log-rank tests for HER2 expression showed no significant association with DFS (log-rank=1.355, p=0.244) or OS (log-rank=0.380, p=0.538), although the 16 HER2 positive patients had a shorter DFS period (Fig. 1).
Figure 1. Kaplan-Meier survival curves. (A) Disease free survival (DFS) curve in 38 patients with HER2 expression. (B) Overall survival (OS) curve in 38 patients with HER2 expression. The HER2 status showed no statistically significant relationship with DFS or OS.
5. Lines 207-212 require references. Please ensure it’s clear which “several studies”.
: In the revised version of the manuscript, we have inserted references. (line 251-255).
6. Lines 235-236 please specify primary serous carcinoma of which organ.
: The indicated sentence has been corrected as “ primary ovarian serous carcinoma” ( line 287 ).
7. The study findings are extremely interesting. Could you add a short paragraph commenting the differences from the results published in the literature?
: In the revised version of the manuscript, we have inserted as shown below (line 334-342).
Previous reports have investigated HER2 expression status in ovarian lesions of HGSOC or regardless of tumor location, but this study focused on peritoneal lesions of HGSOC. HER2 ECD overexpression, HER2 phosphorylation, or various HER2 gene mutations, as well as HER2 ICD overexpression or HER2 gene amplification can lead to HER2 signaling pathway activation. We investigated HER2 status using four different anti-HER2 antibodies. In total, 42.1% (16/38) of peritoneal disseminated HGSOCs showed HER2 positivity with at least one of the four anti-HER2 antibodies or positive HER2 gene amplification.
8. Lines 275-276: can the authors comment on the cost-effectiveness of using multiple HER2 antibodies in clinical practice versus CISH.
: HER2 evaluation in multiple tissue blocks is recommended to accurately assess intratumoral HER2 heterogeneity. Immunohistochemical studies using different HER2 antibodies in multiple tissue blocks can be more cost-effective than in situ hybridazation methods to detect HER2 amplification in multiple blocks.
: In the revised version of the manuscript, we have inserted the above sentences (line 350-353).